# Effect of a Three-Month HOPSports Brain Breaks^®^ Intervention Program on the Physical Fitness Levels of Grade 6-Learners in South Africa

**DOI:** 10.3390/ijerph191811236

**Published:** 2022-09-07

**Authors:** Jacqueline Bonnema, Dané Coetzee, Anita Lennox

**Affiliations:** 1Physical Activity, Sport and Recreation, (PhASRec) Focus Area, Faculty of Health Science, North-West University, Potchefstroom Campus, Private Bag X 6001, Potchefstroom 2531, South Africa; 2School of Management Sciences, North-West University (Vaal Triangle Campus), Vanderbijlpark 1900, South Africa

**Keywords:** physical activity, EUROFIT, children, technology-based intervention, physical education

## Abstract

Despite the numerous health benefits of being physically active, children are not active enough. Various researchers have indicated that intervention programs improve physical fitness levels. Still, only a few have focused on improving physical fitness levels by incorporating technology. HOPSports Brain Breaks^®^ are designed and presented as physical activity solutions with online videos requiring the participants to imitate the movements. These videos are 2–5-min classroom activity breaks. This study determined the effect of a three-month HOPSports Brain Breaks^®^ intervention program on the physical fitness levels of Grade 6-learners. Physical fitness was measured with the EUROFIT test battery. The experimental group consisted of 79 children (26 boys and 47 girls) and the control group of 47 children (16 boys and 33 girls). The mean age for the entire group was 11.92 (±0.36) years. The results indicated that there was a statistically (*p* ≤ 0.05) and practically (*d* ≥ 0.20) significant difference between the experimental and control group for percentage body fat; stork balance; plate tapping; sit-and-reach; standing jump; sit-ups; and 10 × 5 m shuttle run and 20 m shuttle run between the pre-and post-test. Therefore, considering the results mentioned above, the HOPSports Brain Breaks^®^ intervention program can indeed contribute to the improvement of physical fitness, and motor skills of children. Therefore, future studies should be conducted to determine the effect of HOPSports Brain Breaks^®^ between genders as well as what impact it will have on academic performance.

## 1. Introduction

Notwithstanding the numerous health benefits of physical activity (PA) and the role it plays in a child’s life [1,2,3], it is still evident that South African children are not adequately active [1]. Accordingly, the American College of Sports Medicine [4] states that movement skills, aerobic fitness, muscular strength, body composition, cholesterol, blood pressure, blood sugar, and bone health benefit from PA.

Furthermore, research has indicated that PA and physical fitness are beneficial for self-esteem, mood, and character building, positively affecting anxiety [1,5,6]. Various research studies have been conducted which indicate that children’s PA levels increase but reach a peak at the age of 12 years, after which they start to decline [2,7,8]. Recent research concurs and indicates that the prevalence and the decline in PA and physical fitness levels are prevalent and continue until early adulthood. Moreover, it is recommended by researchers that the PA and physical fitness levels of adolescents should be approached differently. Despite the potential success of multi-factor intervention programs, alternative interventions such as digital interventions are needed to address this problem [9].

Research has indicated a positive relationship between physical inactivity and obesity in this sense, negatively impacting children’s physical skills [10,11,12]. In accordance, De Milander [13] studied the relationship between PA, motor proficiency, and physical fitness in 12- to 13-year-old children and reported that lower motor and physical fitness skills result from declining PA levels. Another study by Krombholz [14] indicates that overweight and obese children are, to a lesser extent, physically active and have inferior motor skills. Van Biljon and Longhurst [15] concur and state that decreased physical fitness in obese children could be ascribable to the fact that children do not have the motor skills to perform the activities. In addition, Crane, Naylor, Cook, and Temple [16] stated that as children get older, they are more likely to opt out of PA participation when they do not have the skills to execute the activities. However, children’s confidence and perception of their motor skills could also influence their PA participation, which could improve their physical fitness levels in return [17]. Other factors include lack of resources and facilities, waiting their turn, listening to instructions, and insufficient time in the school curriculum also influence PA and physical fitness [18,19,20,21,22].

The World Health Organization (WHO) [23] defines PA as the ability of the musculoskeletal system to execute a movement, which contributes to energy expenditure. In this sense, having the ability to perform daily PA tasks and leisure-time activities without fatigue summarizes the definition of physical fitness [5,24]. Moreover, physical fitness is categorized into health-related fitness, which is essential to be physically active and preclude chronic diseases [25,26], and sport-related fitness, which includes skills that contribute to the improvement of athletic and sports performance [4,26,27]. Health-related fitness comprises body composition, flexibility, muscular strength and endurance, and cardiovascular endurance components. In contrast, sport-related fitness entails speed, agility, balance, power, reaction time, and coordination [28]. Several researchers have studied the effect of intervention programs on health-related fitness and indicated that intervention programs could improve physical fitness.

Ample research has been done abroad and in South Africa to determine the impact of school-based intervention programs on children’s PA and physical fitness [29,30,31,32,33,34,35,36,37,38]. Moreover, Dallolio et al. [29] studied the impact of a physical education (PE) intervention program and indicated that only the boys’ physical fitness skills improved and recommended that the program should be adapted to meet all the participants’ abilities [29]. In accordance, Šarkauskienė, Derkintienė, and Paplauskas [38] investigated the impact of non-formal PE (e.g., dance, sports, etc.) and found that boys’ cardiovascular fitness, upper body and abdominal strength and endurance, and explosive leg power improved profoundly. In contrast, the girls only indicated improvement in abdominal strength and endurance. Further, Lubans et al. [24] research indicated that older adolescents’ cardiorespiratory and muscular fitness levels improved significantly after a six-month intervention program. The Burn 2 Learn (B2L) program consisted of high-intensity interval training, which teachers implemented during the school’s curriculum. The learners had to complete the activities during their activity breaks, consisting of 20-min multistage fitness tests. Despite this intervention program, the study indicated that the students’ activity and fitness levels decreased when the teachers no longer presented the program [24]. Another study, the ‘Internet-based Professional Learning to help teachers support Activity in Youth’ (iPLAY) program, implemented by Lonsdale et al. [22], reported that physical activity levels, proficiency in movement skills, enjoyment of the sport, and cognitive control improved in the learners. Additionally, Lonsdale et al. [39] also examined the effect of an Internet-based intervention program on improving cardiorespiratory fitness. This study consisted of 22 schools that completed the clinical trial. The program’s content included physical education, sufficient physical activity breaks, and homework the learners needed to complete while physically active. A significant improvement in cardiorespiratory fitness was documented after the 12-month trial. In addition, the study’s results also indicated longevity, as the 24-month assessment showed a sustained improvement in cardiorespiratory fitness. For this reason, programs such as these can be successfully implemented at a population level [39].

Eather [31] investigated the effect of a “fit-for-fun” physical fitness intervention program and found that health-related fitness skills improved, particularly muscular fitness and flexibility. Conversely, Sacchetti et al. [35] studied the efficacy of a school-based intervention program to improve physical skills and performance. It was indicated that the motor performance improved in both groups, whereas the experimental group’s improvements were more significant. Moreover, since children spend most of their time with technology, researchers have recently investigated technology-based intervention programs on physical fitness and PA [11,15,40,41,42,43,44].

Gao [40] investigated the effect of a nine-month Dance Dance Revolution (DDR^®^) program. The results indicated that the experimental group’s daily PA levels increased compared to the control group, whose PA levels declined. In accordance, Tumynaitė et al. [11] studied a three-month HOPSports Brain Breaks^®^ intervention program and its impact on physical fitness and sedentary behavior. It was found that physical fitness did not improve because of the program’s short duration, whereas overall sedentary behavior decreased.

A study on an exergaming intervention program was commenced on 30 overweight and obese children and indicated that physical fitness improved in the experimental group; however, speed, agility, and balance did not improve [15]. Notwithstanding, the HOPSports Brain Breaks^®^ program could contribute to precluding lower PA levels and physical skills and improve physical fitness levels.

In this sense, the literature indicates that more research is necessary to study the effect of HOPSports Brain Breaks^®^ on Grade 6 learners’ physical fitness. This online intervention program can be conducted during class, break, and before and after school to improve PA levels, physical fitness levels, active learning, and test grades [41]. Additionally, this technology-based intervention program incorporates physical, mental, and health videos to increase physical fitness levels and accomplish new motor skills [42].

### Purpose of Research

Literature indicates that the effect of PA and fitness has been studied in detail; however, limited research has been done regarding the impact of technology-based intervention programs such as HOPSports Brain Breaks^®^ on learners’ physical fitness levels. No research could be found on the South African population in this regard. Keeping this in mind, this research aims to study the effect of a three-month HOPSports Brain Breaks^®^ intervention program on the physical fitness levels of Grade 6 learners.

## 2. Materials and Methods

A non-randomized intervention study design was employed for this study with a pre-and post-test. This study was comprised of a convenience sample. This design was chosen to answer the research question: will a HOPSports Brain Breaks^®^ intervention program affect the physical fitness levels of Grade 6 learners in an experimental and control group?

### 2.1. Participants

The focus of this study was to determine the effect of a HOPSports Brain Breaks^®^ intervention program on 114 Grade 6 learners (boys *n* = 56; girls *n* = 58), aged 11- to 12-years. After we obtained a list of all the schools in the Tlokwe Municipality district from the Department of Basic Education, three schools from similar socio-economic backgrounds were randomly selected using the stratified random sampling method. One school functioned as the experimental group (26 boys and 47 girls), and the other two schools served as the control group (16 boys and 33 girls). Please see Figure 1 for more information.

### 2.2. The European Test of Physical Fitness (EUROFIT)

The EUROFIT test was used to determine the physical fitness of the participants. This test is designed for children aged 6- to 18-year-olds and evaluates various fitness components. These components include cardiovascular endurance; running speed and agility; the speed of limb movement; balance; flexibility; explosive leg strength; and abdominal strength, and these were evaluated with the following tests [43]:

20 m (m) shuttle run test: This test assessed cardiovascular endurance and expected the participant to run back and forth between two marks for as long as possible. The test ended when the participant could not finish the lap at a specific time. The number of laps completed was recorded.

10 × 5 m shuttle test: Running speed and agility were evaluated with this test, where the participant had to start behind a line and run 5 m as fast as possible. The participant had to repeat this 10 times; the fastest time was recorded.

Plate tapping test: The plate tapping test evaluated the speed of limb movement. It was expected from the participant to tap two plates with the preferred hand as fast as possible until s/he had performed 25 cycles. The fastest time was recorded.

Stork balance test: The participant’s balance was evaluated with the stork balance test. S/he was expected to stand on one leg for one minute while the non-supporting foot was placed against the medial side of the supporting leg’s knee. The participant’s total time was recorded.

Sit-and-reach test: The sit-and-reach tested hamstring flexibility and required the participant to sit down with bare feet against a sit-and-reach box and keep the knee fully extended. The participant was then asked to reach as far as possible forward. The better of two attempts were recorded to the nearest half centimeter (cm).

Standing long jump test: This test assessed the participant’s explosive leg strength. The participant had to stand behind a line and jump as far as possible with both feet. The most prolonged attempt was recorded.

Sit-up test: The sit-up test was used to determine abdominal strength. The participant had to lie down on their back, knees bent at 90° with the feet flat, and execute as many sit-ups as s/he could in 30 s.

After the completion of each test, the total score was processed into a category, using the EUROFIT reference scales for each skill. Each learner was then allocated into one of the following categories for each skill: (1) Below Average, (2) Average, (3) Above Average, and (4) High Score.

### 2.3. Anthropometry

The measurements were taken by a qualified Level 2 ISAK-accredited anthropometrist according to the standard protocol proposed by ISAK. The following measures were taken of the participants: body mass (kg), stature (m), and skinfolds (triceps, subscapular, and calf). The participants were weighed and measured in light clothing and without shoes, using calibrated scales and a stadiometer. The BMI was calculated as weight (kg)/height (m^2^), and the percentage of body fat was measured by calculating the skinfold measurement using the equation of Slaughter et al. [44].

### 2.4. Intervention

During the intervention program, the control and experimental groups continued participating in the school’s physical education lessons. The control group and experimental group were based in different schools. Both groups were assessed before and after the intervention with the EUROFIT test to determine their physical fitness levels. In addition to the physical education lessons, the experimental group participated in the HOPSports Brain Breaks^®^ intervention program daily for three months (which added up to 12 weeks in total). The researchers and the teachers worked closely to select from the list of various two- to five-minute videos, which included arts (dance and music); fitness skills (cardio and functional fitness); sports (skillastics, cycling, rowing); education (health issues, nutrition, and hygiene); and classroom activities (fun fitness, dynamic physical education) [18]. Each video contains a real or animated instructor to demonstrate the different exercises that would improve children’s motor and physical skills [11,45,46]. The teacher was trained regarding the choice of videos and presentation of the program.

The teachers offered the HOPSports Brain Breaks^®^ intervention program according to the prescribed guidelines for PE as stated in the national curriculum and assessment policy (CAPS) document [47]. Additionally, the program focused on strength, speed, hand-eye and foot-eye coordination, agility, and spatial awareness, among others [18]. For this intervention program, the experimental school had access to the Internet to view the physical activity videos in their classrooms. The participants had to imitate the activities demonstrated in the online video programs, contributing to improved physical fitness and reduced inactivity. It was also expected from the participants to complete the whole video while being physically active. The video intervention program was presented daily in the classroom during school time. There were more than 250 videos in the online video program from which the teachers could choose. The content of the videos includes knowledge about health and training, nutrition, social skills education, environmental education, dance, and fitness [11,18]. Login details and a password were given to the experimental school to gain access to the online video program to present the intervention.

### 2.5. Research Procedure

Consent was obtained through a formal meeting with the corresponding school principals, where the aim and protocol of the study were explained. Informed consent was obtained via letters explaining the aim and protocol of this study to all the Grade 6 learners’ parents and the Grade 6 learners. Participation was entirely voluntary, and they could withdraw from the study anytime. Learners whose parents had given consent were tested to determine their physical fitness, and after that, they followed a HOPSports Brain Breaks^®^ intervention program. The intervention program was done during school time in their classrooms for 12 weeks. The post-test was conducted one week after the intervention program was concluded.

Ethical approval was obtained for this study from the Health Research Ethics Committee of the Faculty of Health Sciences of North-West University (NWU-00003-14-A1). Permission was also obtained from the affiliated schools and the North-West Department of Education. Informed consent was obtained from the parents/legal guardians of the Grade 6 learners before participating in the study. A discussion was held to discuss the purpose of this study and what was expected from each participant whose parent(s) gave consent to participate.

### 2.6. Statistical Analysis

Data processing was done by using SPSS 28.0 (BMI, New York, NY, USA). For descriptive purposes, the data were analyzed by means of arithmetic means (M), standard deviation (SD), and minimum and maximum values. To determine the effect of the HOPSports Brain Breaks^®^ intervention program within and between the experimental and control groups, use was made of independent-sample *t*-test, paired-sample *t*-tests, and ANOVA (repeated measures over-time analysis of variance), as well as a Bonferroni correction for multiple comparisons. The critical level of statistical significance was set at *p* ≤ 0.05, and to determine the practical significance the guidelines of Cohen [48] were used, namely: *d* = 0.2 (small effect), *d* = 0.5 (medium effect) and *d* = 0.8 (large effect).

## 3. Results

Table 1 indicates the composition of the sample population of this study. The experimental group involved 73 participants (26 boys, 47 girls), with a mean age of 11.85 (±0.38), and the control group involved 49 participants (16 boys, 33 girls), with a mean age of 12.02 (±0.31). The entire group of participants was 122, where 42 were boys and 80 were girls. The mean age for the group was 11.92 years (SD = 0.36).

To determine the significance of EUROFIT differences between the groups’ pre-and post-tests, an independent sample *t*-test was conducted. Table 2 reports the differences between the groups regarding the different anthropometry and physical fitness measurements during the pre-and post-test. Regarding the anthropometry measurements in the pre-test, the experimental group was statistically (*p* ≤ 0.05) younger and shorter than the control group compared to the post-test, where height only indicated a practical significance. During the physical fitness tests in the pre-test, the experimental group performed statistically (*p* ≤ 0.05) and practically (*d* ≥ 0.5) lower than the control group regarding the standing long jump (M = 137.29 vs. M = 148.71), sit-ups (M = 15.36 vs. M = 20.59), and the 20 m shuttle run (M = 19.22 vs. M = 25.53), but better during the sit-and-reach test (M = 27.92 vs. M = 22.79). To rectify the differences between the experimental and the control group during the pre-test, a post hoc adjustment was made using the Bonferroni method. Table 2 indicates that after the experimental group participated in the intervention program, the experimental group performed statistically (*p* ≤ 0.05) and practically (*d* ≥ 0.5) better than the control group during the stork balance (*p* ≤ 0.001, *d* ≥ 1.30), plate tapping (*p* ≤ 0.001, *d* ≥ 1.63), sit-and-reach (*p* ≤ 0.05, *d* ≥ 0.73), and 10 × 5 m shuttle run (*p* ≤ 0.05, *d* ≥ 1.13).

To evaluate if any differences occurred within the groups during the pre-and post-tests, paired sample *t*-test was used (see Table 3). Table 3 indicates that during the anthropometric measurements for the experimental group, statistically (*p* ≤ 0.05) and practically (*d* ≥ 0.2) significant differences were observed for the increase in height and body fat percentage, where the experimental group had a higher body fat percentage than the control group. The same tendencies were found for all the physical fitness measurements where the experimental group performed statistically (*p* ≤ 0.05) and practically (*d* ≥ 0.2) better during the post-test after the intervention was received. Regarding the control group, statistically and practically significant differences were also reported regarding the increase in height, weight, BMI, and body fat percentage. However, the control group declined statistically and practically significantly in the post-test during four physical fitness tests (stork balance, plate tapping, standing jump, 10 × 5 m shuttle run, and 20 m shuttle run), but an increase in the number of sit-ups were seen. However, this improvement was not statistically significant, but a practical significance was reported (*d* = 0.5).

A repeated-measures-over-time analysis (ANOVA) was performed (see Table 4) and confirmed the results in Table 2 and Table 3, with a significant group effect regarding the interaction of the two groups over time in all the anthropometric (Table 4; Figure 2a–d) and physical fitness measures (Table 4; Figure 3a–g).

From Table 4 and Figure 2a–d, there were no significant differences between the experimental and control group regarding height (*f* = 1.70; *p* = 0.195), weight ((*f* = 20.61; *p* = 0.232), and BMI (*f* = 1.8; *p* = 0.176), but the body fat percentage (*f* = 7.64; *p* = 0.007) increase significantly more in the experimental group. From Table 4 and Figure 3a–g, it appears that the experimental group reacted significant difference over time if compared to the control group. The intervention effect is visible in Figure 3a–g with regard to differences between the groups in all aspects of the physical fitness skills during the post-test, where the experimental group significantly outperformed the control group (stork balance (*f =* 88.98; *p =* ≤0.001); plate tapping (*f* = 83.97; *p =* ≤0.001); standing long jump (*f* = 32.19; *p =* ≤0.001); sit-ups (*f* = 18.73; *p =* ≤0.001); 10 × 5 m shuttle run (*f* = 60.22; *p =* ≤0.001); 20 m shuttle run (*f* = 381.40; *p =* ≤0.001).

## 4. Discussion

The aim of this study was to determine whether a three-month HOPSports Brain Breaks^®^ intervention program could increase the physical fitness and PA levels of Grade 6-learners.

The experimental group who participated in the HOPSports Brain Break^®^ videos during class and participated in regular PE lessons increased their physical fitness compared to the control group who only participated in regular PE lessons. After the two groups were compared, the results indicated statistically and practically significant differences in the pre- and post-test results. The experimental group reported lower scores for standing jump (leg strength), sit-ups (abdominal strength), and 20 m shuttle run test (cardiovascular endurance) when compared to the control group during the pre-test; however, the experimental group’s sit-and-reach (flexibility) test indicated higher scores than the control group. Accordingly, the results from the post-test indicated that the experimental group performed better in the stork balance (balance), plate tapping (speed), sit-and-reach (flexibility), 10 × 5 m shuttle run (speed and agility), as well as 20 m shuttle run tests (cardiovascular endurance) in comparison to the control group. These results are similar to a study conducted by Tian et al. [33] on 110 children 12- to 13-years old who participated in an enhanced quality PE program. However, these researchers did not make use of the technology-based intervention. The experimental group reported significant differences for sit-and-reach, standing jump, sit-ups, 20 m shuttle run, plate tapping, and 10 × 5 m shuttle run.

Subsequently, the results of our study within each group’s pre- and post-test noted the following differences. The experimental group reported practically and statistically significant improved differences between the pre- and post-test in all the physical fitness components: cardiovascular endurance, abdominal strength, explosive leg power, flexibility, balance, speed, and agility. Various research studies have been done to study the effect of intervention programs on physical fitness; however, only some of the physical fitness components results correlate with the results of this study. In this regard, a study on 113 Grade 1 to Grade 4 learners by Tumynaitė et al. [11] was done where it was expected from the experimental group participated in a three-month HOPSports Brain Breaks^®^ intervention program. The results indicated that boys’ upper body and leg strength improved (*p* ≤ 0.05), whereas agility did not demonstrate any improvement. Furthermore, these results showed that overall physical fitness did not improve; however, PA levels did improve. In another study by Van Biljon and Longhurst [15] on 30 children aged 9- to 12-year-olds, the results indicated that the experimental group who participated in a six-week exergaming intervention program improved their speed, agility, and balance in contrast to the control group who indicated regression in these components. More recent research by Šarkauskienė et al. [38] studied the effect of non-formal PE on 356 Grade 6 learners’ physical fitness. The results reported that boys who participated in the non-formal PE intervention improved their cardiovascular endurance, abdominal strength, and leg strength compared to the group who did not participate in non-formal PE; moreover, no differences in flexibility were indicated. The girls who participated in the non-formal PE only indicated a significant difference in abdominal strength compared to those who did not participate in non-formal PE.

In this study, the control group’s results indicated practical significance for stork balance and sit-ups which had improved after the post-test. In contrast, plate tapping, and 10 × 5 m shuttle run reported both statistical and practical significance lower scores compared to the pre-tests and experimental group’s scores. Standing jump also indicated statistical significance but with lower scores. A possible explanation for the improvement in abdominal strength (sit-up test) can be ascribed to the fact that both boys and girls improve their abdominal strength from the age of 10 years old. In contrast, children from the age of nine years indicated a drastic increase in their balance abilities [28]. In accordance with the results of other research, the control group of this study continued with regular PE classes and might have had limited time to be physically active during the PE period [18,20,21], which could prevent these learners from learning and mastering fundamental skills [15]. However, our control group improved their sit-and-reach scores in comparison to the control group of Stadler [49]. Lastly, it was noted that both groups’ BMI increased in the three months, this could be possibly ascribed to puberty and continues during adolescents [50].

HOPSports Brain Break^®^ video program can enhance PE classes in terms of physical fitness increases [51]. One of the possible explanations for the increase in physical fitness in the experimental group could be that after watching and participating in the various HOPSports Brain Breaks^®^ videos, the learners’ connected the importance and advantages of physical activity with their own PA, which led to an increase in their own PA. In accordance, Bulca et al. [51] concur with this finding that HOPSports Brain Breaks^®^ contributed to the improvement in physical fitness seen in the experimental group and may have had a good impact on students’ overall PA levels.

Therefore, considering the above-mentioned results, the HOPSports Brain Breaks^®^ intervention program can encourage improvements in PA levels and physical fitness for learners and support the PE teachers in teaching learners PA content in an enjoyable manner. Furthermore, overweight, inactive learners or learners with poor motor skills can benefit from this program since it is non-competitive and can be executed at home [51].

The study demonstrated some limitations and should be prevented and acknowledged in future research. However, the study reported valuable outcomes regarding the improvements of physical fitness skills in Grade 6 learners. The main focus of this study was the improvement of the physical fitness levels and not the improvement of physical activity levels. During this study, no gender differences were evaluated. In light of the findings of the study, possible recommendations for future research include determining the effect of this program on the different genders, the effect of a longer intervention period, and the effect of this program on academic performance and how it correlates with PA and fitness. Lastly, it is recommended to focus more on the improvement of PA levels by making use of direct measurements by means of actigraphs.

## 5. Conclusions

Although extensive research has been done to study the effect of various technology-based intervention programs, the findings of the studies vary with little to no improvements. The current study’s results reported that the HOPSports Brain Breaks^®^ intervention program improved all the physical fitness components of the EUROFIT test battery, which could contribute to an increase in PA levels. Therefore, HOPSports Brain Breaks^®^ is an effective program that teachers can incorporate in their PE periods or during class to address the concern of physical inactivity and improve physical fitness skills.

## Figures and Tables

**Figure 1 ijerph-19-11236-f001:**
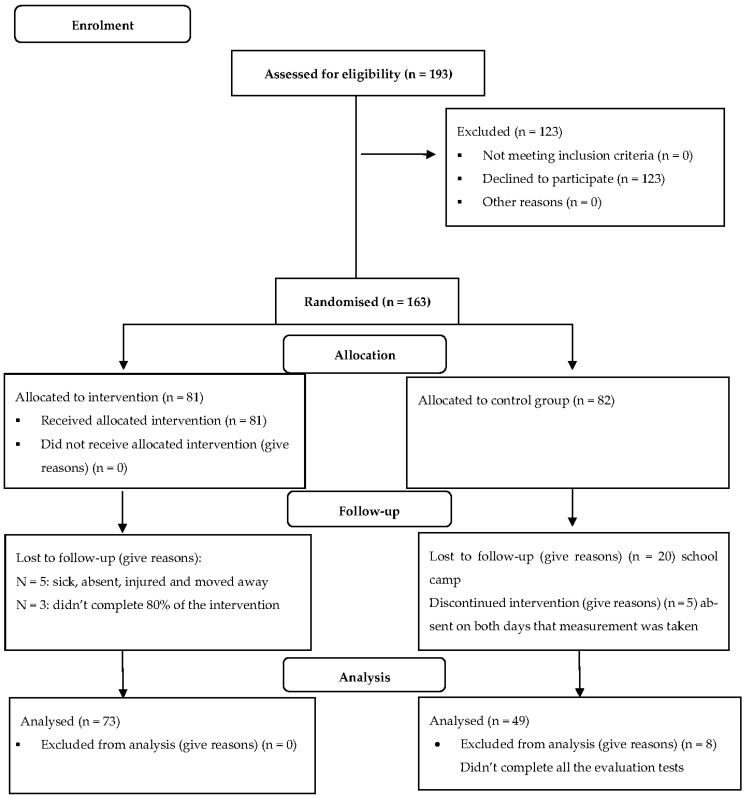
Flow diagram of the participants of this study.

**Figure 2 ijerph-19-11236-f002:**
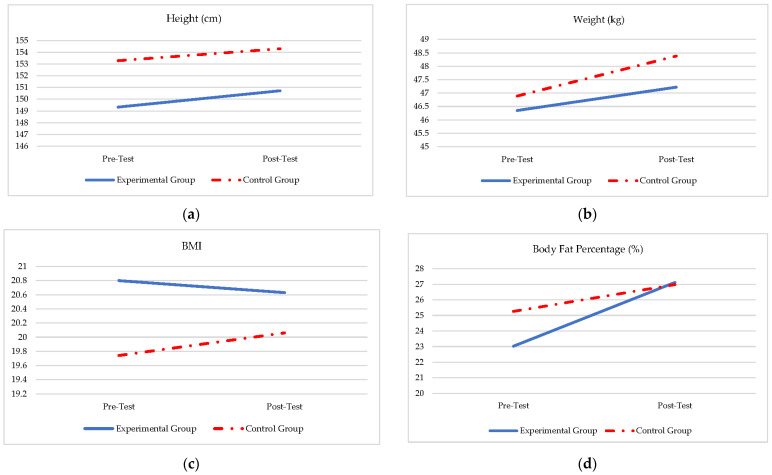
(**a**–**d**)**:** Group interaction over time regarding the anthropometric measurements. (**a**) Group × Time effect for during pre- and post-test (*f* = 1.70; *p* = 0.195). (**b**) Group × Time effect during pre- and post-test (*f* = 20.61; *p* = 0.232). (**c**) Group × Time effect during pre- and post-test (*f* = 1.8; *p* = 0.176). (**d**) Group × Time effect during pre- and post-test (*f* = 7.64; *p* = 0.007).

**Figure 3 ijerph-19-11236-f003:**
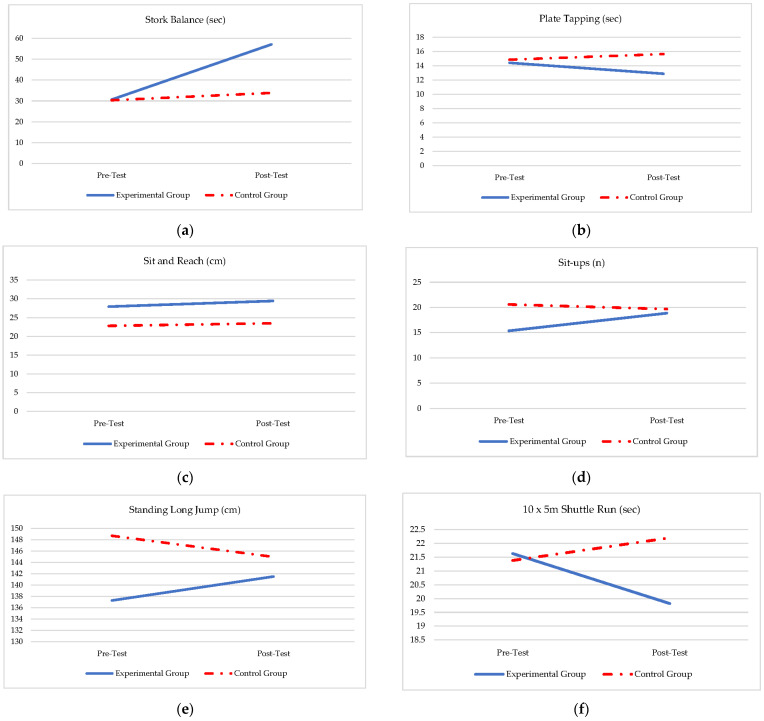
(**a**–**g**). Group interaction over time regarding physical fitness skills. (**a**) Group × Time effect during pre- and post-test (*f =* 88.98; *p =* ≤0.001); (**b**) Group × Time effect during pre- and post-test (*f* = 83.97; *p =* ≤0.001); (**c**) Group × Time effect during pre- and post-test (*f* = 0.52; *p* = 0.473); (**d**) Group × Time effect during pre- and post-test (*f* = 32.19; *p =* ≤0.001); (**e**) Group × Time effect during pre- and post-test (*f* = 18.73; *p =* ≤0.001); (**f**) Group × Time effect during pre- and post-test (*f* = 60.22; *p =* ≤0.001); (**g**) Group × Time effect during pre- and post-test (*f* = 381.40; *p =* ≤0.001).

**Table 1 ijerph-19-11236-t001:** Descriptive statistics of study variables.

Subjects	N	Min	Max	Mean	SD
Experimental group	73	10.50	13.02	11.85	0.38
Control group	49	11.42	12.70	12.02	0.31
Total group	122	10.50	13.02	11.92	0.36

N = number, Min = minimum; Max = maximum; SD = standard deviation.

**Table 2 ijerph-19-11236-t002:** Differences between the experimental and control group during the pre- and post-test regarding their physical fitness levels.

Variables	Experimental Group (*n* = 73)	Control Group (*n* = 49)	Significance of Difference			
	Mean ± SD	Mean ± SD	*t*	df	*p*	*d*
Pre-Test (PrT)						
Anthropometry						
Age (years)	11.85 ± 0.38	12.02 ± 0.31	−2.786	114.305	<0.006 *	0.47 ^#^
Height (cm)	149.33 ± 6.95	153.28 ± 8.29	−2.745	90.528	<0.007 *	0.48 ^#^
Weight (kg)	46.35 ± 16.10	46.89 ± 11.91	−0.212	118.809	0.832	0.03
BMI	20.80 ± 6.33	19.74 ± 3.78	1.152	118.631	0.252	0.17
% Body fat	23.02 ± 8.30	25.26 ± 7.20	−1.580	112.244	0.117	0.27 ^#^
Physical Fitness						
Stork balance (sec)	30.62 ± 14.96	30.33 ± 16.82	0.099	94.700	0.921	0.02
Plate tapping (sec)	14.42 ± 2.06	14.88 ± 1.44	−1.466	119.790	0.145	0.23 ^#^
Sit-and-reach (cm)	27.92 ± 7.32	22.79 ± 8.00	3.594	96.769	<0.001 *	0.64 ^##^
Standing jump (cm)	137.29 ± 21.40	148.71 ± 19.65	−3.037	108.860	<0.003 *	0.53 ^##^
Sit-up (n)	15.36 ± 4.24	20.59 ± 3.14	−7.824	118.768	<0.001 *	1.23 ^#^
10 × 5 m Shuttle run (sec)	21.63 ± 2.69	21.37 ± 1.48	0.699	116.307	0.486	0.10
20 m Shuttle run (n)	19.22 ± 11.34	25.53 ± 12.69	−2.810	95.052	<0.006 *	0.50 ^##^
Post-Test (PoT)						
Anthropometry						
Height (cm)	150.72 ± 7.01	154.30 ± 8.44	−2.456	89.859	0.016	0.42 ^#^
Weight (kg)	47.22 ± 16.44	48.38 ± 12.86	−0.438	117.134	0.662	0.07
BMI	20.63 ± 6.08	20.06 ± 4.17	0.613	119.926	0.541	0.09
% Body fat	27.12 ± 9.42	26.98 ± 7.99	0.087	113.482	0.931	0.01
Physical Fitness						
Stork balance (sec)	57.08 ± 6.45	33.83 ± 17.83	8.749	56.528	<0.001 *	1.30 ^###^
Plate tapping (sec)	12.89 ± 1.69	15.64 ± 1.59	−9.141	107.365	<0.001 *	1.63 ^###^
Sit-and-reach (cm)	29.43 ± 7.14	23.47 ± 8.20	4.140	93.204	<0.001 *	0.73 ^##^
Standing jump (cm)	141.52 ± 19.69	144.98 ± 19.71	−0.951	103.013	0.344	0.18
Sit-up (n)	18.86 ± 4.61	19.67 ± 8.69	−0.598	66.280	0.552	0.09
10 × 5 m Shuttle run (sec)	19.82 ± 2.11	22.20 ± 1.88	−6.512	110.775	<0.001 *	1.13 ^###^
20 m Shuttle run (n)	28.85 ± 15.93	24.82 ± 12.93	1.537	115.605	0.127	0.25 ^#^

SD = standard deviation; df = degrees of freedom; cm—centimeters; kg—kilogram; %—percentage; sec—seconds; n—number; *t* = *t*-value; * statistical significance *p* ≤ 0.05; ^#^ practical significance small effect *d* = ≥0.2; ^##^ practical significance medium effect *d* = ≥0.5; ^###^ practical significance large effect, *d* = ≥0.8.

**Table 3 ijerph-19-11236-t003:** Differences within the experimental and control group’s physical fitness levels during pre-and post-test.

	Pre-Test (PrT)	Post-Test (PoT)	Significance of Difference			
Variables	Mean ± SD	Mean ± SD	*t*	df	*p*	*d*
Experimental group						
Anthropometry						
Height (cm)	149.33 ± 6.95	150.72 ± 7.01	−6.939	72	<0.000 *	0.20 ^#^
Weight (kg)	46.35 ± 16.10	47.22 ± 16.44	−2.229	72	0.029	0.05
BMI	20.80 ± 6.33	20.63 ± 6.08	0.597	72	0.552	0.03
% Body fat	23.02 ± 8.30	27.12 ± 9.42	−6.622	72	<0.001 *	0.49 ^#^
Physical Fitness						
Stork balance (sec)	30.62 ± 14.96	57.08 ± 6.45	−14.818	72	<0.001 *	1.77 ^##^
Plate tapping (sec)	14.42 ± 2.06	12.89 ± 1.69	8.757	72	<0.001 *	0.74 ^##^
Sit-and-reach (cm)	27.92 ± 7.32	29.43 ± 7.14	−2.970	72	<0.004 *	0.21 ^#^
Standing jump (cm)	137.29 ± 21.40	141.52 ± 19.69	−4.867	72	<0.001 *	0.20 ^#^
Sit-up (n)	15.36 ± 4.24	18.86 ± 4.61	−12.343	72	<0.001 *	0.83 ^###^
10 × 5 m Shuttle run (sec)	21.63 ± 2.69	19.82 ± 2.11	8.175	72	<0.001 *	0.68 ^##^
20 m Shuttle run (n)	19.22 ± 11.34	28.85 ± 15.93	−10.500	72	<0.001 *	0.85 ^###^
Control group						
Anthropometry						
Height (cm)	153.28 ± 8.29	154.30 ± 8.44	−6.229	48	<0.001 *	0.12
Weight (kg)	46.89 ± 11.91	48.38 ± 12.86	−5.828	48	<0.001 *	0.13
BMI	19.74 ± 3.78	20.06 ± 4.17	−2.658	48	0.011 *	0.08
% Body fat	25.26 ± 7.20	26.98 ± 7.99	−3.483	48	<0.001 *	0.24 ^#^
Physical Fitness						
Stork balance (sec)	30.33 ± 16.82	33.83 ± 17.83	2.425	48	0.010 *	0.21 ^#^
Plate tapping (sec)	14.88 ± 1.44	15.64 ± 1.59	−2.665	48	<0.001 *	0.52 ^##^
Sit-and-reach (cm)	22.79 ± 8.00	23.47 ± 8.20	−4.798	48	0.566	0.08
Standing jump (cm)	148.71 ± 19.65	144.98 ± 19.71	−0.577	48	<0.002 *	0.19
Sit-up (n)	15.36 ± 4.24	19.67 ± 8.69	3.327	48	0.439	0.50 ^##^
10 × 5 m Shuttle run (sec)	21.37 ± 1.48	22.20 ± 1.88	0.781	48	<0.002 *	0.56 ^##^
20 m Shuttle run (n)	25.53 ± 12.69	24.82 ± 12.93	−3.290	48	0.019 *	0.06

SD = standard deviation; df = degrees of freedom; cm—centimeters; kg—kilogram; %—percentage; sec—seconds; n—number; *t* = *t*-value; * *p* ≤ 0.05; ^#^ practical significance small effect *d* = ≥0.2; ^##^ practical significance medium effect *d* = ≥0.5; ^###^ practical significance large effect, *d* = ≥0.8.

**Table 4 ijerph-19-11236-t004:** Effect of the intervention program on the experimental- and control group regarding the anthropometric measurements and physical fitness tests.

Variable	df	Mean^2^	F-Value	*p*-Value
Anthropometry				
Height (cm)				
Interaction effect	1	5,412,709.99	47,751.24	≤0.001 *
Time effect	1	84.96	74.77	≤0.001 *
Time × Group effect	1	1.93	1.70	0.195
Group effect	1	830.83	7.33	0.008 *
Weight (kg)				
Interaction effect	1	522,763.20	1197.359	≤0.001 *
Time effect	1	81.81	20.61	≤0.001 *
Time × Group effect	1	5.73	1.45	0.232
Group effect	1	42.55	0.097	0.755
BMI				
Interaction effect	1	96,723.64	1697.21	≤0.001 *
Time effect	1	0.34	0.178	0.674
Time × Group effect	1	3.49	1.853	0.176
Group effect	1	38.70	0.679	0.412
Body fat %				
Interaction effect	1	153,667.08	1182.04	≤0.001 *
Time effect	1	498.01	46.07	≤0.001 *
Time × Group effect	1	82.60	7.641	0.007 *
Group effect	1	64.570	0.497	0.482
Physical Fitness				
Stork Balance (sec)				
Interaction effect	1	338,076.53	1081.20	≤0.001 *
Time effect	1	13,157.58	151.67	≤0.001 *
Time × Group effect	1	7719.56	88.98	≤0.001 *
Group effect	1	8119.99	25.97	≤0.001 *
Plate Tapping (sec)				
Interaction effect	1	49,027.79	9459.45	≤0.001 *
Time effect	1	8.92	9.76	0.002 *
Time × Group effect	1	76.76	83.97	≤0.001 *
Group effect	1	151.67	29.26	≤0.001 *
Sit-and-Reach (cm)				
Interaction effect	1	157,355.18	1636.97	≤0.001 *
Time effect	1	69.85	3.640	0.059
Time × Group effect	1	9.95	0.518	0.473
Group effect	1	1804.18	18.77	≤0.001 *
Standing Long Jump (cm)				
Interaction effect	1	4,804,888.09	6095.46	≤0.001 *
Time effect	1	3.64	0.126	0.723
Time × Group effect	1	930.64	32.190	≤0.001 *
Group effect	1	3248.37	4.121	0.045
Sit-ups (n)				
Interaction effect	1	81,331.87	1917.066	≤0.001 *
Time effect	1	98.22	6.410	0.013 *
Time × Group effect	1	287.08	18.733	≤0.001 *
Group effect	1	535.90	12.632	0.001 *
10 × 5 m Shuttle Run (sec)				
Interaction effect	1	105,944.76	13,932.109	≤0.001 *
Time effect	1	14.23	8.37	0.005 *
Time × Group effect	1	102.46	60.22	≤0.001 *
Group effect	1	65.48	8.611	0.004 *
20 m Shuttle Run (n)				
Interaction effect	1	141,989.26	415.955	≤0.001 *
Time effect	1	1165.35	60.47	≤0.001 *
Time × Group effect	1	1568.71	81.403	≤0.001 *
Group effect	1	76.10	0.223	0.638

df—degrees of freedom; cm—centimeters; kg—kilogram; %—percentage; sec—seconds; n—number; *p*—Greenhouse–Geisser; *p* ≤ 0.05 *.

## Data Availability

The dataset is the property of the North-West University under the supervision of Dané Coetzee. In this regard, D. Coetzee should be contacted if, for any reason, the data included in this paper need to be shared. D. Coetzee is the principal investigator of this study and gave permission that the data can be used.

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
