# Peer review of "Effect of a Three-Month HOPSports Brain Breaks® Intervention Program on the Physical Fitness Levels of Grade 6-Learners in South Africa"

_ijerph, 2022, doi:10.3390/ijerph191811236_

Round 1
Reviewer 1 Report
Manuscript Review July 2022
Manuscript Number: ijerph-1816647
Title: Effect of a three-month HOPSports Brain Breaks® intervention programme on the physical fitness levels of Grade 6-learners in South Africa
Abstract
Page 1
Line 12: Change ‘there is still’ to ‘there are still’. Also, change ‘that is not’ to ‘that are not’.
Lines 15 and 16: Add 1 sentence for the abstract on what the HOPSports Brain Breaks® intervention programme is i.e. more details warranted.
Lines 17 and 18: Re-write this sentence on gender breakdown – in its current form, it reads grammatically off.
Line 24: Remove the words ‘with different motor abilities’.
Line 27: Replace the keywords that are already repeated in the manuscript title – please avoid repetition.
Overall, the abstract will need some revisions structurally and grammatically in the next submission..
Introduction
Page 1
Line 39: Change ‘between inactivity’ to ‘between physical inactivity’. Also insert comma to read ‘between physical inactivity and obesity, which negatively…’.
Line 42: Change ‘lower motor’ to ‘lower motor skills’. Also, change ‘physical fitness skills is a result’ to ‘physical fitness are a result’.
Page 2
Lines 35 to 54: There is too much happening within this introductory paragraph. I would strongly suggest removing the volume of text to ensure that the reader is accessing succinct and precise information within the section.
Line 58: Change ‘active is’ to ‘active are’.
Lines 59 and 60: Why are there 3 x references for defining physical activity – this is unnecessary and complicates the sentence.
Lines 62 and 36: Please re-write the sentence beginning with ‘In this sense,…’ - in its current form, it reads grammatically off.
Lines 55 to 71: Again, there is too much happening within this introductory paragraph. I would strongly suggest removing the volume of text to ensure that the reader is accessing succinct and precise information within the section. For example, in this paragraph, the reader hears about wellbeing, fundamental movement skills, health related fitness and physical fitness – this is very hard to process in terms of paragraph flow.
Line 80: There is an extra space between the end of the sentence and the next sentence.
Lines 89 to 92: Please re-write the sentence beginning with ‘It was found,…’ - in its current form, it reads grammatically off.
Line 96: Remove the words ‘who did not improve’.
Page 3
Line 113: Consider replacing ‘nine’ with the number ‘9’.
Line 117: Remove the word ‘skill’.
Line 119: Remove the words ‘who indicated only improvement in flexibility’.
Lines 72 to 121: These sentences relate to intervention programmes, however, similar to my comments above, there is too much introductory text and the introduction overall needs to be cleaned more precisely and succinctly.
Line 127: Change words to read ‘contribute to the increase of PA levels…’.
Introduction Summary:
Overall, the introduction is text heavy and a little bit disjointed in terms of the arguments being raised. As mentioned already, there is too much introductory text and the introduction overall needs to be cleaned more precisely and succinctly. I do believe that less text and shorter arguments will enhance the flow of the section – I would advise a member of the authorship team with English fluency to proof-read this revised section.
Material and Methods
Page 3
Line 137: Change sentence to read ‘This study comprised of a convenience sample…’.
Line 147: How were schools allocated to either the experimental or control group condition? Please expand.
Page 4
Line 149: Table 1 can be deleted, it is unnecessarily repetitive to what has been reported in the text already.
Line 153: Consider replacing ‘six’ with the number ‘6’.
Line 156: Comment on the validity and reliability of the EUROFIT test battery for the population in question.
Line 196: Are the words ‘fitness skills’ grammatically correct?
Page 5
Lines 202 to 203: What do the words ‘presented within the framework for PE provided in the CAPS’ mean? Please clarify and explain this sentence further.
Line 208: Remove the words ‘during the intervention’.
Line 209: Change the words ‘they can choose’ to ‘participants could choose’.
Line 212: What is unclear to me at this point is the following: where did the experimental school practice, participate and engage in the daily intervention? How did they participate in the intervention? More practical details warranted.
Line 233: Change ‘each test variables’ to ‘each of the test variables’.
Line 234: Using a t-test over time seems off statistically speaking? Did you control for any additional variables?
Methods and Methods Summary
I commend the authors’ for providing a text-based overview on the data collection measures, alongside the intervention programme content. The data analysis procedures undertaken might need to be considered further, specifically to control for additional variables which might have influenced participants physical fitness performances over time. Some specific sentences as outlined above will need some further clarity and detail.
Results
Page 6
Lines 251 to 253: Change sentence to read: ‘The experimental group involved 73 participants (26 boys, 47 girls), with a mean age of 11.85 (±0.38), and the control group involved 49 participants (16 boys, 33 girls), with a mean age of 12.02 (±0.31).’
Lines 253 to 254: Change sentence to read: ‘The total group of participants was 122, where…’.
Line 261: Re-write to read ‘… post-tests, independent sample t-tests were conducted.’.
Line 262: Change ‘fitness’ to ‘physical fitness’.
Line 267: Change ‘worse’ to ‘lower’. Also rephrase sentence to ‘regarding the standing long…’.
Line 271: Change ‘indicate’ to ‘indicates’.
Line 279: When you say: ‘differences were observed for the increase in height and body fat percentage’, who are you talking about exactly? What groups? More clarity and specificity warranted.
Lines 279 to 282: Is the following sentence actually true: ‘The same tendencies were found for all the physical fitness measurements where the experimental group performed statistically (p≤0.05) and practically (d≥0.2) better during the post-test after the intervention was received.’ Please look specifically at the plate tapping findings – perhaps I am missing something here though, because it looks like the experimental group’s performance in plate tapping disimproved over time?
Pages 8 to 12:
I am very unsure as too where figures 1 to 8 come from within the purpose of the current manuscript. Are these 8 figures actually needed? At no stage in the introduction to this manuscript or within the methods section do the authorship team refer to the differing EUROFIT classification bands. Was the investigation of low scores to high scores across the differing physical fitness parameters part of your research question? As a reader, these specific findings on pages 8 to 12 come as a surprise to me, as I thought the study’s objective was to examiner the effectiveness (or not) of the intervention programme over time, when compared to the control group. These findings need to be better framed within the manuscript or else, they need to be removed. Please watch an oversaturation of figures also, and consider changing the y axis to the percentage values.
Results Summary
Overall, the data presentation for the pre and post comparisons across the physical fitness and anthropometric variables are clear initially and aligned to your research question. Please see my comment above in terms of pages 8 to 12 though – in its current form, I cannot see how these specific results fit within your manuscript. The overall results and justification of data analysis will need a significant edit to ensure that the manuscript focus is clear from the offset. There should be no surprises for the reader within the data that is analysed and presented.
Discussion
Page 13
Line 437: Change ‘will’ to ‘could’.
Line 437: Was PA measured or reported in the results? I apologise, I seem to have missed this outcome within the text. Please clarify, thank you.
Lines 450 to 451: I am not seeing critical discussion on how the current study used technology within the intervention programme. This discussion point is not obvious or clear to me from reading the manuscript i.e. ‘although these researchers didn’t make use of technology-based intervention.’
Lines 454 to 476: There is a conscious effort from the authors to flag somewhat comparable intervention programme findings, however, what do the current results mean in terms of the manuscript’s contribution to the field. Specifically, what do the increases in physical fitness measures at post-test for the experimental group in the intervention programme mean. How and why did these physical fitness performances increase so significantly for the experimental group over 3 months?
Lines 477 to 484: It is unusual to see the control group being isolated within a paragraph discussion as a standalone write-up. I would have expected more specific information towards the brain breaks intervention programme being discussed.
Page 14
Lines 485 to 498: The latter part of this discussion paragraph is more focused and specific to the research questions (e.g. lines 498 to 503). The opening to this paragraph discusses the classification of the EUROFIT data categories, however, as mentioned in the results, this type of analysis did not feature in the introduction and/or the materials/methods sections.
Line 505: Why PA levels? How can you quantify this? You make this argument much more convincing on line 512.
Line 513: You conclude with PE periods and class time, however, the reader is not made aware of these practicalities in terms of the intervention delivery within the manuscript. Further work is warranted here.
Line 514: Change ‘inactivity’ to ‘physical inactivity’.
Discussion Summary:
Overall, this section commences with some critically insightful discussion points in the context of other international physical fitness intervention programme findings. That being said, the reasons as to why such findings were observed are not aligned to and/or discussed in the context of the brain breaks intervention programme offering. This is an oversight of the discussion section, as the authors spend more time repeating the findings without clear and synergistic links to the brain breaks intervention content and programme. I suggest that the authors do need to expand further on some of the intervention pillars and the practicalities of this intervention in school settings.
Author Response
Good afternoon
Thank you for the time you took to review our article and made suggestions to improve our article.
Kind regards
Dané
|
Rebuttal letter for article ijerph-1816647: Reviewer 1
|
|
|
Reviewer 1 |
|
|
Abstract |
|
|
Line 12: Change ‘there is still’ to ‘there are still’. Also, change ‘that is not’ to ‘that are not’. |
Revised according to recommendation |
|
Lines 15 and 16: Add 1 sentence for the abstract on what the HOPSports Brain Breaks® intervention programme is i.e. more details warranted. |
Revised according to recommendation |
|
Lines 17 and 18: Re-write this sentence on gender breakdown – in its current form, it reads grammatically off. |
Revised according to recommendation |
|
Line 24: Remove the words ‘with different motor abilities’. |
Revised according to recommendation |
|
Line 27: Replace the keywords that are already repeated in the manuscript title – please avoid repetition. |
Revised according to recommendation |
|
Overall, the abstract will need some revisions structurally and grammatically in the next submission.. |
Revised according to recommendation |
|
Introduction |
|
|
Line 39: Change ‘between inactivity’ to ‘between physical inactivity’. Also insert comma to read ‘between physical inactivity and obesity, which negatively…’. |
Revised according to recommendation |
|
Line 42: Change ‘lower motor’ to ‘lower motor skills’. Also, change ‘physical fitness skills is a result’ to ‘physical fitness are a result’. |
Revised according to recommendation |
|
Lines 35 to 54: There is too much happening within this introductory paragraph. I would strongly suggest removing the volume of text to ensure that the reader is accessing succinct and precise information within the section. |
Revised according to recommendation |
|
Line 58: Change ‘active is’ to ‘active are’. |
Revised according to recommendation |
|
Lines 59 and 60: Why are there 3 x references for defining physical activity – this is unnecessary and complicates the sentence. |
Revised according to recommendation |
|
Lines 62 and 36: Please re-write the sentence beginning with ‘In this sense,…’ - in its current form, it reads grammatically off. |
Revised according to recommendation |
|
Lines 55 to 71: Again, there is too much happening within this introductory paragraph. I would strongly suggest removing the volume of text to ensure that the reader is accessing succinct and precise information within the section. For example, in this paragraph, the reader hears about wellbeing, fundamental movement skills, health related fitness and physical fitness – this is very hard to process in terms of paragraph flow. |
Revised according to recommendation |
|
Line 80: There is an extra space between the end of the sentence and the next sentence. |
Revised according to recommendation |
|
Lines 89 to 92: Please re-write the sentence beginning with ‘It was found,…’ - in its current form, it reads grammatically off. |
Revised according to recommendation |
|
Line 96: Remove the words ‘who did not improve’. |
Revised according to recommendation |
|
Line 113: Consider replacing ‘nine’ with the number ‘9’. |
Revised according to recommendation |
|
Line 117: Remove the word ‘skill’. |
Revised according to recommendation |
|
Line 119: Remove the words ‘who indicated only improvement in flexibility’. |
Revised according to recommendation |
|
Lines 72 to 121: These sentences relate to intervention programmes; however, similar to my comments above, there is too much introductory text and the introduction overall needs to be cleaned more precisely and succinctly. |
Revised according to recommendation |
|
Line 127: Change words to read ‘contribute to the increase of PA levels…’. |
Revised according to recommendation |
|
Introduction Summary: Overall, the introduction is text heavy and a little bit disjointed in terms of the arguments being raised. As mentioned already, there is too much introductory text and the introduction overall needs to be cleaned more precisely and succinctly. I do believe that less text and shorter arguments will enhance the flow of the section – I would advise a member of the authorship team with English fluency to proof-read this revised section. |
Revised according to recommendation Recent articles have been added Overall grammar and English fluency have been revised |
|
Materials and Methods |
|
|
Line 137: Change sentence to read ‘This study comprised of a convenience sample…’. |
Revised according to recommendation |
|
Line 147: How were schools allocated to either the experimental or control group condition? Please expand. |
Revised according to recommendation |
|
Line 149: Table 1 can be deleted, it is unnecessarily repetitive to what has been reported in the text already. |
Revised according to recommendation |
|
Line 153: Consider replacing ‘six’ with the number ‘6’. |
Revised according to recommendation |
|
Line 196: Are the words ‘fitness skills’ grammatically correct? |
Revised according to recommendation |
|
Lines 202 to 203: What do the words ‘presented within the framework for PE provided in the CAPS’ mean? Please clarify and explain this sentence further. |
Revised according to recommendation |
|
Line 208: Remove the words ‘during the intervention’. |
Revised according to recommendation |
|
Line 209: Change the words ‘they can choose’ to ‘participants could choose’. |
Revised according to recommendation |
|
Line 209: Change the words ‘they can choose’ to ‘participants could choose’. |
Revised according to recommendation |
|
Line 212: What is unclear to me at this point is the following: where did the experimental school practice, participate and engage in the daily intervention? How did they participate in the intervention? More practical details warranted. |
Revised according to recommendation |
|
Line 233: Change ‘each test variables’ to ‘each of the test variables’. |
Revised according to recommendation |
|
Line 234: Using a t-test over time seems off statistically speaking? Did you control for any additional variables? |
Revised according to recommendation |
|
Methods and Methods Summary |
|
|
I commend the authors’ for providing a text-based overview on the data collection measures, alongside the intervention programme content. The data analysis procedures undertaken might need to be considered further, specifically to control for additional variables which might have influenced participants physical fitness performances over time. Some specific sentences as outlined above will need some further clarity and detail. |
Revised according to recommendations. Did consult with statistical services of the North-West University |
|
Results |
|
|
Lines 251 to 253: Change sentence to read: ‘The experimental group involved 73 participants (26 boys, 47 girls), with a mean age of 11.85 (±0.38), and the control group involved 49 participants (16 boys, 33 girls), with a mean age of 12.02 (±0.31).’ |
Revised according to recommendation |
|
Lines 253 to 254: Change sentence to read: ‘The total group of participants was 122, where…’. |
Revised according to recommendation |
|
Line 261: Re-write to read ‘… post-tests, independent sample t-tests were conducted.’. |
Revised according to recommendation |
|
Line 262: Change ‘fitness’ to ‘physical fitness’. |
Revised according to recommendation |
|
Line 267: Change ‘worse’ to ‘lower’. Also rephrase sentence to ‘regarding the standing long…’ |
Revised according to recommendation |
|
Line 271: Change ‘indicate’ to ‘indicates’. |
Revised according to recommendation |
|
Line 279: When you say: ‘differences were observed for the increase in height and body fat percentage’, who are you talking about exactly? What groups? More clarity and specificity warranted. |
Revised according to recommendation |
|
Lines 279 to 282: Is the following sentence actually true: ‘The same tendencies were found for all the physical fitness measurements where the experimental group performed statistically (p≤0.05) and practically (d≥0.2) better during the post-test after the intervention was received.’ Please look specifically at the plate tapping findings – perhaps I am missing something here though, because it looks like the experimental group’s performance in plate tapping disimproved over time? |
“The plate tapping test evaluated the speed of limb movement. It was expected from the participant to tap two plates with the preferred hand as fast as possible until s/he had performed 25 cycles. The fastest time was recorded.” The time did improve because the learners executed the activity faster. |
|
Pages 8 to 12: |
|
|
I am very unsure as too where figures 1 to 8 come from within the purpose of the current manuscript. Are these 8 figures actually needed? At no stage in the introduction to this manuscript or within the methods section do the authorship team refer to the differing EUROFIT classification bands. Was the investigation of low scores to high scores across the differing physical fitness parameters part of your research question? As a reader, these specific findings on pages 8 to 12 come as a surprise to me, as I thought the study’s objective was to examiner the effectiveness (or not) of the intervention programme over time, when compared to the control group. These findings need to be better framed within the manuscript or else, they need to be removed. Please watch an oversaturation of figures also, and consider changing the y axis to the percentage values. |
Revised according to recommendation |
|
Results Summary |
|
|
Overall, the data presentation for the pre and post comparisons across the physical fitness and anthropometric variables are clear initially and aligned to your research question. Please see my comment above in terms of pages 8 to 12 though – in its current form, I cannot see how these specific results fit within your manuscript. The overall results and justification of data analysis will need a significant edit to ensure that the manuscript focus is clear from the offset. There should be no surprises for the reader within the data that is analysed and presented. |
Revised according to recommendation |
|
Discussion |
|
|
Line 437: Change ‘will’ to ‘could’. |
Revised according to recommendation |
|
Line 437: Was PA measured or reported in the results? I apologise, I seem to have missed this outcome within the text. Please clarify, thank you. |
Revised according to recommendation |
|
Lines 450 to 451: I am not seeing critical discussion on how the current study used technology within the intervention programme. This discussion point is not obvious or clear to me from reading the manuscript i.e. ‘although these researchers didn’t make use of technology-based intervention.’ |
Revised according to recommendation |
|
Lines 454 to 476: There is a conscious effort from the authors to flag somewhat comparable intervention programme findings, however, what do the current results mean in terms of the manuscript’s contribution to the field. Specifically, what do the increases in physical fitness measures at post-test for the experimental group in the intervention programme mean. How and why did these physical fitness performances increase so significantly for the experimental group over 3 months? |
Revised according to recommendation |
|
Lines 477 to 484: It is unusual to see the control group being isolated within a paragraph discussion as a standalone write-up. I would have expected more specific information towards the brain breaks intervention programme being discussed. |
Revised according to recommendation |
|
Lines 485 to 498: The latter part of this discussion paragraph is more focused and specific to the research questions (e.g. lines 498 to 503). The opening to this paragraph discusses the classification of the EUROFIT data categories, however, as mentioned in the results, this type of analysis did not feature in the introduction and/or the materials/methods sections. |
Revised according to recommendation |
|
Line 505: Why PA levels? How can you quantify this? You make this argument much more convincing on line 512. |
Revised according to recommendation |
|
Line 513: You conclude with PE periods and class time, however, the reader is not made aware of these practicalities in terms of the intervention delivery within the manuscript. Further work is warranted here.
|
Revised according to recommendation |
|
Line 514: Change ‘inactivity’ to ‘physical inactivity’. |
Revised according to recommendation |
|
Discussion Summary: |
|
|
Overall, this section commences with some critically insightful discussion points in the context of other international physical fitness intervention programme findings. That being said, the reasons as to why such findings were observed are not aligned to and/or discussed in the context of the brain breaks intervention programme offering. This is an oversight of the discussion section, as the authors spend more time repeating the findings without clear and synergistic links to the brain breaks intervention content and programme. I suggest that the authors do need to expand further on some of the intervention pillars and the practicalities of this intervention in school settings. |
Revised according to recommendation |
Reviewer 2 Report
Overal comments:
I would like to praise the authors for conducting this study. The topic is needed, and the interventional approach seems promising. Authors reported a serious of positive effects on fitness and anthropometric components due to an online intervention based on video activities.
Despite the valuable research question and interventional approach, I have some serious concerns on the conceptual basis of the study as well as the interventional design and description. Therefore, I am only commenting on the introduction and methods section for this version.
Specific comments:
Introduction:
I encourage authors to revise the structure of the introduction. Currently, it is too extensive and deviates from the main focus of the paper which was to test an intervention program delivered via online videos to improve fitness level and antrhopometrics.
Authors include a series of observational studies, especially cross-sectional and miss the most up-to-date references in the field.
For example, these three REFs conducted large scaled interventions and were not cited in the introduction:
1. doi: 10.1136/bjsports-2020-103277
2. doi: 10.1186/S12889-016-3243-2
3. doi: 10.1001/jamapediatrics.2021.0417
This systematic review (doi: 10.1007/S40279-021-01516-8) summarises the most recent publications around the Stodden et al's model that the authors briefly mentioned in the introduction. This other paper, suggests an extension of the Stodden et al.'s model (doi: 10.1007/s40279-021-01632-5). In all the abovementioned papers, authors can find the most recent publications in the field.
Authors need to present the rational of their intervention programme in their introduction, which is not clearly presented in addition to the added value of this work in comparison to the current state of the art.
Line 38, REFs 2,7,8: There is more updated REF on this statement - please revise and rephrase the statement accordingly.
doi:10.1016/S0140-6736(21)01259-9
Methods:
Line 137-8: Please provide a citation for this study design since I have never heard of this design. Other reader must also be unfamiliar.
How the intervention was designed? Please expand its description.
Intervention: how this videos were supposed/expected to be implemented in daily practice? Does the authors have any indicator of the adherence of the intervention?
Statistical analysis: ANOVA is required for the main analysis. Since this is not a RCT, authors must consider to adjust for possible confounders, thus linear regression would be needed. It is necessary to evaluate whether intervention and control groups differed at baseline. Currently, this is not described in the statistical analysis.
Author Response
Good afternoon
Thank you for the time you took to review our article and made suggestions to improve our article.
Kind regards
Dané
|
Rebuttal letter for article ijerph-1816647: Reviewer 2
|
|
|
Reviewer 2 |
|
|
Introduction |
|
|
I encourage authors to revise the structure of the introduction. Currently, it is too extensive and deviates from the main focus of the paper which was to test an intervention program delivered via online videos to improve fitness level and anthropometrics. |
Revised according to the recommendation Overall grammar and English fluency revised More recent articles were added |
|
Authors include a series of observational studies, especially cross-sectional, and miss the most up-to-date references in the field. |
Revised according to the recommendation |
|
For example, these three REFs conducted large scaled interventions and were not cited in the introduction: |
Revised according to the recommendation |
|
1. doi: 10.1136/bjsports-2020-103277 |
Revised according to the recommendation Recent articles have been added. |
|
2. doi: 10.1186/S12889-016-3243-2 |
Revised according to the recommendation Recent articles have been added. |
|
3. doi: 10.1001/jamapediatrics.2021.0417 |
Revised according to the recommendation Recent articles have been added. |
|
This systematic review (doi: 10.1007/S40279-021-01516-8) summarises the most recent publications around the Stodden et al's model that the authors briefly mentioned in the introduction. This other paper, suggests an extension of the Stodden et al.'s model (doi: 10.1007/s40279-021-01632-5). In all the above mentioned papers, authors can find the most recent publications in the field. |
Revised according to the recommendation |
|
Authors need to present the rational of their intervention programme in their introduction, which is not clearly presented in addition to the added value of this work in comparison to the current state of the art. |
Revised according to the recommendation |
|
Line 38, REFs 2,7,8: There is more updated REF on this statement - please revise and rephrase the statement accordingly. |
Revised according to the recommendation |
|
doi:10.1016/S0140-6736(21)01259-9 |
Revised according to the recommendation Recent articles have been added. |
|
Materials and Methods |
|
|
Line 137-8: Please provide a citation for this study design since I have never heard of this design. Other reader must also be unfamiliar. |
Revised according to the recommendation |
|
How the intervention was designed? Please expand its description. |
Revised according to the recommendation |
|
Intervention: how these videos were supposed/expected to be implemented in daily practice? Does the authors have any indicator of the adherence of the intervention? |
Revised according to the recommendation |
|
Statistical analysis: ANOVA is required for the main analysis. Since this is not a RCT, authors must consider to adjust for possible confounders, thus linear regression would be needed. It is necessary to evaluate whether intervention and control groups differed at baseline. Currently, this is not described in the statistical analysis. |
Revised according to recommendations. Did consult with statistical services of the North-West University |
Reviewer 3 Report
The study makes a new contribution to the effectiveness of this intervention programme that has already been demonstrated in the literature. I would suggest to the authors to expand the description of the HOPSports Brain Breaks® intervention programme, as it is not very explanatory. The activities carried out, and the overall duration, should be more detailed. The positive impacts attributable to this intervention programme should also be further detailed
Author Response
Good afternoon
Thank you for the time you took to review our article and made suggestions to improve our article.
Kind regards
Dané
|
Rebuttal letter for article ijerph-1816647: Reviewer 3
|
|
|
Reviewer 3 |
|
|
Introduction |
|
|
The study makes a new contribution to the effectiveness of this intervention programme that has already been demonstrated in the literature. I would suggest to the authors to expand the description of the HOPSports Brain Breaks® intervention programme, as it is not very explanatory. The activities carried out, and the overall duration, should be more detailed. The positive impacts attributable to this intervention programme should also be further detailed
|
Revised according to the recommendation Overall grammar and English fluency revised More recent articles were added |